# The Natural Product Parthenolide Inhibits Both Angiogenesis and Invasiveness and Improves Gemcitabine Resistance by Suppressing Nuclear Factor κB Activation in Pancreatic Cancer Cell Lines

**DOI:** 10.3390/nu16050705

**Published:** 2024-02-29

**Authors:** Yuki Denda, Yoichi Matsuo, Saburo Sugita, Yuki Eguchi, Keisuke Nonoyama, Hiromichi Murase, Tomokatsu Kato, Hiroyuki Imafuji, Kenta Saito, Mamoru Morimoto, Ryo Ogawa, Hiroki Takahashi, Akira Mitsui, Masahiro Kimura, Shuji Takiguchi

**Affiliations:** Department of Gastroenterological Surgery, Nagoya City University Graduate School of Medical Sciences, 1-Kawasumi, Mizuho-cho, Mizuho-ku, Nagoya 467-8601, Japan; denda@med.nagoya-cu.ac.jp (Y.D.); ssabu3753@gmail.com (S.S.); yukieg0802@gmail.com (Y.E.); knonoyama0924@gmail.com (K.N.); muramen5.com@gmail.com (H.M.); tomo.k.g.w@gmail.com (T.K.); himafuji1979@gmail.com (H.I.); kentaxis777@gmail.com (K.S.); morimamo1121@gmail.com (M.M.); ryogawancu@gmail.com (R.O.); coolsound1230@gmail.com (H.T.); a.mitsui.21@west-med.jp (A.M.); m.kimura@med.nagoya-cu.ac.jp (M.K.); takiguch@med.nagoya-cu.ac.jp (S.T.)

**Keywords:** parthenolide, pancreatic cancer, gemcitabine resistance, NF-κB, invasion, angiogenesis

## Abstract

We previously established pancreatic cancer (PaCa) cell lines resistant to gemcitabine and found that the activity of nuclear factor κB (NF-κB) was enhanced upon the acquisition of gemcitabine resistance. Parthenolide, the main active ingredient in feverfew, has been reported to exhibit antitumor activity by suppressing the NF-κB signaling pathway in several types of cancers. However, the antitumor effect of parthenolide on gemcitabine-resistant PaCa has not been elucidated. Here, we confirmed that parthenolide significantly inhibits the proliferation of both gemcitabine-resistant and normal PaCa cells at concentrations of 10 µM and higher, and that the NF-κB activity is significantly inhibited, even by 1 µM parthenolide. In Matrigel invasion assays and angiogenesis assays, the invasive and angiogenic potentials were higher in gemcitabine-resistant than normal PaCa cells and were inhibited by a low concentration of parthenolide. Furthermore, Western blotting showed suppressed MRP1 expression in gemcitabine-resistant PaCa treated with a low parthenolide concentration. In a colony formation assay, the addition of 1 µM parthenolide improved the sensitivity of gemcitabine-resistant PaCa cell lines to gemcitabine. These results suggest that parthenolide may be used as a novel therapeutic agent for the treatment of gemcitabine-resistant PaCa.

## 1. Introduction

Pancreatic cancer (PaCa) has one of the poorest prognoses of all cancers as well as high mortality rates. In 2021, in Japan [1], EU countries [2], and the U.S. [3], PaCa was the fourth, fourth, and third most common cause of death among all cancer-related deaths, respectively. In 2022, in the U.S., more than 64,000 new cases of PaCa were diagnosed, and approximately 50,000 patients died from PaCa [4]. Furthermore, it is estimated that the total incidence of PaCa in the U.S. will increase by 30% by 2040 [5]. Approximately half of PaCa patients have distant metastases upon presentation, with a 5-year relative survival rate of only 3.2% [4]. Although there have been many studies on chemotherapy for PaCa, and attempts have been made to develop drugs based on those studies [6,7,8,9], there are still very few treatment options for PaCa compared with other carcinomas. In 1997, a phase III trial was conducted in PaCa patients that compared fluorouracil monotherapy, the standard of care at the time, with gemcitabine (Gem) monotherapy. The trial demonstrated the superiority of Gem over fluorouracil in overall survival (5.7 vs. 4.4 months) [10]. As a result, Gem was approved as the first-line chemotherapeutic agent for locally advanced or metastatic PaCa, and it remains the standard of care for PaCa [11]. In 2013, a randomized controlled trial (Prep-02/JSAP05 trial) was initiated to compare the results of upfront surgery versus two preoperative courses of Gem plus S-1 therapy. The median overall survival was significantly improved by the preoperative therapy compared with surgery alone (26.6 vs. 36.7 months) [12]. However, while the trial results were promising, clinical experience has shown that, although Gem chemotherapy acts transiently on PaCa initially, its effect readily diminishes thereafter. The development of Gem resistance is, thus, a major clinical hurdle, but its mechanism has not been elucidated. The clarification of the Gem resistance mechanism will hopefully lead to improvements in PaCa treatments.

Previously, we established PaCa cell lines with acquired Gem resistance [13]. We then showed that the acquisition of Gem resistance enhanced the activity of NF-κB [14]. The NF-κB transcription factor is involved in the regulation of cell proliferation, apoptosis, and inflammatory responses; the stimulation of invasion and metastasis; and the promotion of cancer development and its progression via multiple mechanisms [15]. Several inflammatory factors, such as IL-1 and TNF-α, are regulated by NF-κB [16], yet they are also potent inducers of NF-κB; in addition, these factors are strongly associated with most types of PaCa [17]. Furthermore, many chemotherapeutic agents, such as Gem, have been shown to activate NF-κB in PaCa cells [18]. Thus, targeting NF-κB signaling pathways simultaneously with the use of chemotherapeutic agents may be a better option for PaCa treatment [19,20]. 

Parthenolide is a sesquiterpene lactone derived from the leaves of the medicinal plant *Tanacetum parthenium* [21]. Parthenolide has been used in herbal medicine for centuries for its anti-inflammatory and anti-migraine properties [22,23,24]. Recently, the anticancer properties of parthenolide have received significant attention. Parthenolide inhibits the activity of the upstream inhibitor of the NF-κB (IκB) kinase (IKK) complex, preventing the degradation of two NF-κB regulatory proteins, IκB-α and IκB-β, and inhibiting NF-κB activity [25]. Parthenolide has already shown efficacy against various types of human cancer cells, including acute myeloid leukemia, chronic myeloid leukemia, prostate cancer, cervical cancer, and hepatocellular carcinoma cells [26,27,28,29,30,31]. However, whether parthenolide suppresses activated NF-κB signaling or exerts antitumor effects in Gem-resistant PaCa cells has not been reported.

The aim of this study was to determine whether parthenolide has an antitumor effect on Gem-resistant PaCa cells by suppressing NF-κB activity.

## 2. Materials and Methods

### 2.1. Reagents

Parthenolide was purchased from AdipoGen Life Sciences (cat. No. AG-CN2-0455; Lausanne, Switzerland). A 10 mM parthenolide solution was prepared in DMSO (Sigma-Aldrich, St. Louis, MO, USA), stored in small aliquots at 4 °C. Recombinant human tumor necrosis factor (TNF) α was purchased from R&D Systems Inc. (cat. No. 210-TA; Minneapolis, MN, USA).

### 2.2. Cell Lines and Cell Culture

The AsPC-1 (cat. CRL-1682) and MIA PaCa-2 (cat. CRL-1420) and EA.hy926 (cat. CRL-2922) were purchased from the American Type Culture Collection (Manassas, VA, USA). The cells were cultured as previously described [13,32]. The media used were Roswell Park Memorial Institute medium 1640 (cat. QJ-R8758; Sigma-Aldrich; Merck KGaA, Darmstadt, Germany) and Dulbecco’s modified Eagle’s medium (DMEM; cat. QJ-D6429; Sigma-Aldrich; Merck KGaA). Both media were supplemented with 10% fetal bovine serum (FBS), penicillin (10,000 U/mL), amphotericin B (25 µg), and streptomycin (10 mg/mL) (all from Gibco, Grand Island, CA, USA). 

### 2.3. Establishment of PaCa Cell Lines Resistant to Gem

Gem-resistant (GR) pancreatic cancer cell lines were established as described in detail previously [13]. Gem was purchased from Eli Lilly Japan K.K. (Kanagawa, Japan). The half-maximal inhibitory concentration (IC_50_) of Gem was determined by constructing a dose–response curve for each PaCa strain using WST-1 assay (cat. No. MK400; Takara Bio, Yamanashi, Japan). Briefly, when passaging each PaCa cell line, Gem was added at a concentration consistent with the IC_50_, and the cells were passaged repeatedly for 2–3 weeks. After that, the IC_50_ of Gem for each cell line was measured again, a new concentration of Gem was added and cells were passaged repeatedly, and this procedure was repeated. Gem resistance was defined as a cell line exhibiting a IC_50_ of Gem at least a 50-fold greater than the parental cell lines.

### 2.4. Cytotoxicity Assay

The cytotoxicity of parthenolide was assessed using the WST-1 assay (cat. No. MK400; Takara Bio) according to the manufacturer’s protocol. Briefly, Gem-sensitive (GS) and GR AsPC-1 or GS and GR MIA PaCa-2 cells were seeded at 3 × 10^3^/well in 100 µL in 96-well plates and incubated with various concentrations of parthenolide (0–100 µM) for 48 h. A SpectraMax ABS microplate reader (Molecular Devices, San Jose, CA, USA) was used to measure the assay. 

### 2.5. Immunocytochemical Analysis of NF-κB p65 Localization

Immunocytochemical analysis was performed, with some modifications mentioned in our previous study [32]. GS and GR AsPC-1 cells or GS and GR MIA PaCa-2 cells were seeded in four-chamber glass slides (1 × 10^4^/chamber) and cultured overnight. The pretreatment time for parthenolide (1 μM) was 1 h, and the stimulation time for TNF-α (0.5 ng/mL) was 15 min. Untreated cells were used as controls. The cells were fixed in 4% paraformaldehyde and permeabilized with 0.1% Triton-X. The cells were incubated with anti-NF-κB p65 antibodies (cat. No. 8242; Cell Signaling Technology, Danvers, MA, USA) at 1:400 dilutions firstly and then with fluorescent secondary antibodies (cat. No. ab150077; Abcam, Cambridge, UK) at 1:500 dilutions. The nuclei were visualized using DAPI staining (cat. No. P36931; Thermo Fisher Scientific, Rockford, IL, USA). Fluorescence images were captured using a BZ-X710 fluorescent microscope (Keyence Corporation, Osaka, Japan).

### 2.6. Nuclear Protein Extraction and NF-κB p65 Activity Assays 

NF-κB activity was measured using a total of 10 μg of nuclear extract according to the manufacturer’s protocol for the Trans AM NF-κB p65 kit (catalog No. 40096; Active Motif, Inc.). The pretreatment of parthenolide (1 µM) and TNF-α (0.5 ng/mL) was provided for 1 h and 15 min, respectively. Nuclear extracts were collected using the Nuclear Extraction kit (cat. No. 40010; Active Motif, Inc., Carlsbad, CA, USA) according to the manufacturer’s instructions and normalized using a BCA assay (cat. No. 23225; Thermo Fisher Scientific). 

### 2.7. Western Blotting of Multidrug Resistance Protein 1 (MRP1)

Western blotting was performed using an iBind Flex Solution kit (Thermo Fisher Scientific, Rockford, IL, USA), as previously described [33]. Each cell was treated with or without parthenolide (1 µM) in 10% FBS for 24 h and whole proteins were extracted using a radioimmunoprecipitation lysis buffer supplemented with the Protease and Phosphatase Inhibitor Cocktail (Thermo Fisher Scientific). The protein samples were quantified using a BCA assay (cat. No. 23225; Thermo Fisher Scientific). Each protein extract (20 µg per lane) was separated on 10% SDS-PAGE (Bio-Rad Laboratories, Inc., Hayward, CA, USA) and transferred to nitrocellulose membranes. The membranes were probed with primary and secondary antibodies. The primary antibodies were anti-MRP1 (cat. No. 72202S; Cell Signaling Technology, Inc.) at 1:1000 dilutions and anti-GAPDH (cat. No. 2118S; Cell Signaling Technology) at 1:2000 dilutions. The immunoreactive protein bands were detected using an Amersham Imager 600 (Cytiva, Uppsala, Sweden). The ImageJ software, v1.53 (National Cancer Institute, Bethesda, MD, USA), was used to conduct a densiometric analysis of the bands from the blots.

### 2.8. ELISA

GS and GR AsPC-1 cells or GS and GR MIA PaCa-2 cells were seeded in 6-well plates (1 × 10^5^/well), containing the appropriate medium supplemented with 10% FBS, and incubated overnight at 37 °C. Each well was then replaced with FBS-free medium with or without parthenolide (1 µM) and TNF-α (0.5 ng/mL) and incubated for 48 h. Analogous to a previous study [32], the concentrations of IL-8 and VEGF in the supernatant were determined using the ELISA kit (cat. No. D8000C and DVE00; R&D Systems, Inc., Minneapolis, MN, USA).

### 2.9. Matrigel Invasion Assay

The invasion ability of PaCa was verified using BioCoat Matrigel Invasion Chambers (Corning, Inc., Corning, NY, USA) with some modifications from previous studies [13,33]. Each cell was seeded (1 × 10^5^/chamber) in the upper chamber with the appropriate medium without FBS. In addition, TNF-α (0.5 ng/mL) and parthenolide (1 µM) were added to the upper and lower chambers. Then, 10% FBS was added to the lower chamber as a chemoattractant. The invaded cells were stained using the Diff-Quick cell-staining kit (cat. No. ZS1009; Sysmex Co., Kobe, Japan), and counted in nine random microscopic fields (magnification, ×200).

### 2.10. Tube Formation Assay for Angiogenesis

In order to assess the angiogenesis ability, a tube formation assay was performed using the EA.hy926 cell line and a Matrigel Matrix (cat. No. 354230, Corning, Inc.), as previously described [32,33]. Briefly, GS and GR AsPC-1 cells or GS and GR MIA PaCa-2 cells were seeded in 6-well plates (1 × 10^5^/well) containing the appropriate medium with 10% FBS. After overnight incubation at 37 °C, the medium was changed and the cells were incubated for an additional 48 h in the presence or absence of parthenolide (1 μM) and TNF-α (0.5 ng/mL) in 2% FBS. The supernatant was centrifuged and used as a conditioning medium. DMEM and the above supernatant were mixed in equal volumes, added to wells coated Matrigel together with EA.hy926 cells (1.2 × 10^4^/well), and incubated for 16 h. The formation of capillary-like structures was observed using the BZ-X710 fluorescence microscope (Keyence Corporation), and the number of intersections was counted.

### 2.11. Colony Formation Assay

The colony formation assay was performed as follows. Briefly, single-cell suspensions of GS and GR AsPC-1 cells or GS and GR MIA PaCa-2 cells were prepared, seeded in 60 mm dishes (500 cells/dish), and incubated overnight at 37 °C. Then, the medium in the dishes was replaced with fresh medium containing various concentrations of Gem (0–100 µM) with and without parthenolide (1 µM), and the cells were incubated for 10 days. Finally, the cells were stained with Diff-Quick (cat. No. ZS1009; Sysmex Co.), and the visible colonies were counted manually. A population consisting of 50 or more cells was defined as a colony.

### 2.12. Statistical Analysis

Data from experiments performed three or more times are presented as the mean ± standard deviation, an unpaired *t*-test and one-way ANOVA and Bonferroni’s post hoc test were used to compare differences between groups using the software, EZR v1.61 (Saitama Medical Center, Jichi Medical University, Saitama, Japan), a graphical user interface for R (The R Foundation for Statistical Computing). *p* < 0.05 was considered statistically significant.

## 3. Results

### 3.1. IC_50_ Values of Gem and Parthenolide in PaCa Cell Lines

First, to measure the IC_50_ values of Gem and parthenolide in our established Gem-resistant cell line, we performed WST-1 assays on two PaCa cell lines: AsPC-1 (GR and GS) and MIA PaCa-2 (GR and GS). After adding various concentrations of Gem or parthenolide and incubating for 48 h, the effect of Gem or parthenolide on cell proliferation was examined using a WST-1 assay. The IC_50_ values of Gem after 48 h of treatment were 0.045, 229.5, 0.051, and 28.64 µM for GS AsPC-1, GR AsPC-1, GS MIA PaCa-2, and GR MIA PaCa-2, respectively (Figure 1). The IC_50_ values of parthenolide after 48 h of treatment were 2.53, 2.72, 2.94, and 3.47 µM for GS AsPC-1, GR AsPC-1, GS MIA PaCa-2, and GR MIA PaCa-2, respectively (Figure 2). To avoid a cytotoxic effect from the parthenolide, the parthenolide concentration used in subsequent experiments was 1 µM, which was lower than the IC_50_.

### 3.2. PaCa Increases p65 Activity Induced by Gem Resistance, While Parthenolide Inhibits p65 Activity by Blocking the Nuclear Translocation of p65

First, an immunocytochemical analysis was performed to determine whether parthenolide affected the nuclear translocation of p65 in each PaCa cell (Figure 2a). In all the PaCa cells, treatment with TNF-α alone caused p65 to translocate to the nucleus; however, treatment with a low concentration of parthenolide (1 µM) inhibited the nuclear translocation, and p65 remained in the cytoplasm. Next, the nuclear proteins were extracted, and the amount of p65 translocation to the nucleus was measured using a Trans AM NF-κB p65 kit (Figure 2b). The activity of p65 can be determined using this kit by measuring the amount of p65 that moves into the nucleus and binds to p65 binding sites. The results showed that the p65 activity was more enhanced by Gem resistance in both cell lines, and that parthenolide significantly reduced the p65 activity. In AsPC-1 GS and GR cells, treatment with parthenolide decreased the p65 activity by 71.5% and 85.9%, respectively, compared to the control, and treatment with parthenolide also decreased the p65 activity, which was increased by TNF-α, by 51.8% and 85.9%, respectively. The p65 activities of MIA PaCa-2 GS and GR cells were decreased by 61.6% and 47.9%, respectively, by the parthenolide treatment compared to the control, and the p65 activities increased by TNF-α were also decreased by 71.7% and 16.5%, respectively, by the parthenolide treatment. These results suggest that parthenolide exerts an antitumor effect by suppressing the p65 activity enhanced by Gem resistance. 

### 3.3. The Invasion Ability of PaCa Is Increased by Gem Resistance, but Suppressed by Parthenolide

A Matrigel invasion assay was performed to determine the effect of parthenolide on the invasion ability of each cell line. The GR PaCa cell lines (AsPC-1 and MIA PaCa-2) had a more enhanced cell invasion ability compared with the GS PaCa cell lines. The addition of parthenolide significantly inhibited the invasive ability in all cell lines. Additionally, the addition of TNF-α enhanced the invasion of PaCa cells, but this effect was significantly suppressed by a low concentration of parthenolide (1 μM) (Figure 3). The specific rate of the decrease in the invasion ability due to parthenolide is shown below. AsPC-1 GS and GR reduced the number of invasive cells by 64.5% and 31.2%, respectively, compared with the control, and suppressed the invasive ability increased by TNF-α by 73.5% and 63.5%, respectively; MIA PaCa-2 GS and GR also reduced the number of invasive cells by 61.9% and 27.6%, respectively, compared with the control, and suppressed the invasive ability increased by TNF-α by 67.6% and 31.9%, respectively.

### 3.4. Angiogenesis of PaCa Is Increased by Gem Resistance, but Suppressed by Parthenolide

The effect of parthenolide on the tube-forming ability in human endothelial cells was evaluated. The group incubated with the supernatant of GR PaCa cells had more enhanced tube formation compared to the group of GS PaCa cells. Furthermore, tube formation was significantly enhanced versus suppressed after incubation with the supernatant of PaCa cells treated with TNF-α versus those treated with parthenolide, respectively. In addition, the tube formation promoted by TNF-α was significantly inhibited by treatment with parthenolide (Figure 4). The specific reduction rate of tube formation by parthenolide is shown below. AsPC-1 GS and GR decreased tube formation by 24.9% and 36.6%, respectively, compared with the control, and TNF-α-enhanced tube formation was also suppressed by 13.1% and 15.7%, respectively. MIA PaCa-2 GS and GR also decreased tube formation by 34.7% and 30.5% compared with the control, respectively, and TNF-α-enhanced tube formation was also suppressed by 32.0% and 19.9%, respectively. 

### 3.5. Parthenolide Suppresses the Secretion of IL-8 and VEGF from PaCa Cells

Changes in the amounts of the angiogenic factors IL-8 and VEGF secreted from PaCa cells were evaluated using ELISA. The secreted levels of IL-8 and VEGF were increased by Gem resistance. Furthermore, the addition of TNF-α enhanced the VEGF and IL-8 protein secretion in all the PaCa cell lines, which was significantly inhibited by the addition of parthenolide. The secretion ability of both proteins, which was enhanced by TNF-α, was also suppressed by 1 μM parthenolide (Figure 5). The specific reduction rate of the IL-8 and VEGF secretion ability by parthenolide is shown below. AsPC-1 GS and GR decreased IL-8 secretion by 14.9% and 39.5%, respectively, and VEGF secretion by 56.2% and 38.2%, respectively, compared with the control; the enhancement of the IL-8 secretion by TNF-α was also suppressed by 22.6% and 25.1%, respectively, and the VEGF secretion was decreased by 38.8% and 35.9%, respectively. In MIA PaCa-2 GS and GR, the IL-8 secretory capacity was reduced by 39.7% and 57.0%, respectively, and the VEGF secretory capacity was reduced by 59.1% and 44.6%, respectively, compared with the control; the IL-8 secretion enhanced by TNF-α was also suppressed by 40.6% and 33.0%, respectively, and the VEGF secretion was decreased by 44.2% and 31.8%, respectively.

### 3.6. Parthenolide Suppresses the Expression of the Multidrug-Resistance-Related Protein MRP1 and Improves the Sensitivity to Gem

Lastly, to investigate whether parthenolide can improve Gem sensitivity, we examined the expression of the multidrug-resistance-related protein MRP1 using Western blotting. MRP1 was overexpressed in Gem-resistant PaCa cells compared with normal PaCa cells, and this expression was suppressed by 1 µM parthenolide (Figure 6a–d). Gem-resistant PaCa cells were then treated with a combination of 1 µM parthenolide and Gem and evaluated in a colony formation assay. The colony-forming ability of the GR AsPC-1 and MIA PaCa-2 cells was significantly suppressed by the parthenolide-plus-Gem combination therapy compared with Gem monotherapy (Figure 6e,f).

## 4. Discussion

The aim of this study was to clarify whether parthenolide has an antitumor effect on PaCa by inhibiting the NF-κB transcription factor, the activity of which is increased upon the acquisition of Gem resistance in PaCa cells. We found that parthenolide inhibited NF-κB activity and suppressed the invasiveness and angiogenesis in two different human PaCa cell lines. Furthermore, parthenolide suppressed the invasiveness and angiogenesis of Gem-resistant PaCa cells. The expression of MRP1, which is thought to be involved in multidrug resistance, was also decreased by parthenolide treatment, suggesting that parthenolide may also improve Gem sensitivity. To our knowledge, this is the first research to focus on enhanced NF-κB activity in Gem-resistant PaCa and to show associations of parthenolide with the suppressed invasive potential, the production of angiogenic factors such as VEGF and IL-8, and an improved Gem sensitivity via the suppression of MRP1 expression.

NF-κB has been implicated in apoptosis, invasion, metastasis, and angiogenesis [34,35]. Inactivated NF-κB forms a complex with IκB in the cytoplasm; the activation of NF-κB results in the phosphorylation and ubiquitination of IκB, which is eventually degraded by the proteome. The free NF-κB then migrates into the nucleus and induces transcription [36]. TNF-α is one activator of NF-κB [37]. In this study, we established Gem-resistant PaCa cells and evaluated the nuclear proteins; the results showed that NF-κB activity was enhanced by Gem resistance, and NF-κB was activated by TNF-α and suppressed by parthenolide. Parthenolide also sufficiently inhibited GR cells, in which NF-κB activity is increased. Furthermore, fluorescence immunostaining suggested that parthenolide suppresses the activity of NF-κB by inhibiting its nuclear translocation.

The relevance of NF-κB to the invasive potential of PaCa has been discussed in previous studies [14,38]. Consistent with previous results, NF-κB regulation altered the invasive potential of GS PaCa cells. NF-κB was regulated upward in GR PaCa cells compared to GS PaCa cells, and invasive capacity was likewise enhanced in GR PaCa cells. Furthermore, the invasion ability was enhanced by TNF-α treatment in both GS and GR PaCa cells. The inhibition of NF-κB by parthenolide sufficiently inhibited GR PaCa cells and also inhibited the invasive potential enhanced by TNF-α.

Next, we evaluated the angiogenic potential of GR PaCa cells. The progressive growth and metastasis of PaCa are thought to depend on the tumor’s invasive potential and angiogenic factors released from stromal cells [39]. VEGF and IL-8, among a number of angiogenic factors, are considered important mediators of angiogenesis in PaCa [40,41]. Our previous studies have shown that PaCa cell lines with a higher liver metastasis capacity secrete more IL-8 [42], and that IL-8 is linked to angiogenesis via the IL-8/CXCR2 axis in PaCa [38,43]. Moreover, NF-κB regulates the secretion of VEGF and IL-8 in PaCa, and NF-κB suppression has been shown to inhibit angiogenesis by suppressing the production of VEGF and IL-8 [44]. It has also been demonstrated that PaCa enhances tumor angiogenesis by increasing IL-8 production via the acquisition of Gem resistance [13]. Although PaCa generally presents as an ischemic tumor, several studies have shown an association between the microvessel density and the PaCa prognosis [45,46,47]. Indeed, several studies have demonstrated the efficacy of angiogenesis inhibition in PaCa [48,49]. Furthermore, our research group previously reported that novel therapies targeting angiogenesis inhibit tumor enlargement in PaCa in vivo [43,50].

Therefore, we investigated whether parthenolide inhibits angiogenesis in human endothelial cells. The results of this study showed that culturing EA.hy926 cells with the supernatant of PaCa cells, which appears to contain VEGF and IL-8, enhances the angiogenic potential, while culturing with the supernatant of PaCa cells treated with parthenolide, which appears to contain less VEGF and IL-8, inhibits the angiogenic potential. In fact, when these secretion levels were measured using ELISA, the supernatants of the PaCa cells treated with parthenolide showed significantly lower VEGF and IL-8 levels than those of the untreated PaCa cells. In summary, parthenolide inhibited NF-κB, suppressed the VEGF and IL-8 secretion from PaCa cells, and inhibited angiogenesis by EA.hy926 cells. Furthermore, parthenolide had a marked inhibitory effect on GR PaCa cells, reducing the VEGF and IL-8 secretion and angiogenic activity.

NF-κB may be involved in chemotherapy resistance [51]. One factor that contributes to resistance to Gem, a key drug in PaCa therapy, is the ATP-binding cassette (ABC) transporter [52]. These transporters actively expel drugs from cancer cells, reducing their effectiveness and contributing to multidrug resistance [53]. The ABC transporter superfamily has seven subfamilies (A–G) with a total of 48 members and is a specialized class of proteins located on the plasma membrane [54]. Previous reports have shown that ABC transporters play an important role in Gem resistance in PaCa [55]. Among them, MRP1 contributes to the development of multidrug resistance [56]. In fact, MRP1 was suggested to be highly expressed in PaCa tissues and cells resistant to Gem [57]. Furthermore, a high expression level of MRP1 is involved in Gem efflux, and a high MRP1 expression level can cause Gem resistance [58]. In addition, the upregulation of MRP1 in pancreatic tumors may be responsible for a poor therapeutic response in general and a shortened postoperative survival [57]. A relationship between MRP1 expression and gemcitabine sensitivity has been reported in liver cancer [59], cholangiocarcinoma [60], and lung cancer [52], suggesting a correlation between gemcitabine resistance and increased MRP1 expression. Based on the above, we focused on MRP1 as a mechanism of gemcitabine resistance in pancreatic cancer. In this study, both PaCa cell lines showed an increased expression of MRP1 in GR cells, and parthenolide significantly decreased the MRP1 expression in GR PaCa cells and increased the Gem sensitivity. This suggests that MRP1 expression is positively correlated with NF-κB, but further verification of the signaling pathway is needed.

Our study found that PaCa enhanced the invasive and angiogenic potential by Gem resistance via increased NF-κB activation. A low concentration of parthenolide inhibited the invasive potential by suppressing NF-κB activation in GR PaCa cells and inhibited angiogenesis by suppressing the production of VEGF and IL-8. Furthermore, parthenolide enhanced Gem sensitivity by suppressing MRP1 expression, which contributed to the development of multidrug resistance. Parthenolide at a low concentration showed antitumor effects without affecting cell growth, suggesting that it may be safer and less toxic than existing anticancer drugs. Therefore, parthenolide is expected to be an effective therapeutic agent against GR PaCa, but further studies, such as in vivo experiments (e.g., animal experiments using nude mice), are needed before it can be used in clinical practice. 

## 5. Conclusions

In conclusion, GR PaCa cells have stronger invasive and angiogenic ability compared with GS cells, and they may achieve resistance to Gem via enhanced NF-κB signaling. A low concentration of parthenolide inhibited the invasive and angiogenic potential in GS and GR PaCa cell lines and also increased the Gem sensitivity of GR cells. Thus, parthenolide may be an effective treatment option for PaCa patients who have developed Gem resistance.

## Figures and Tables

**Figure 1 nutrients-16-00705-f001:**
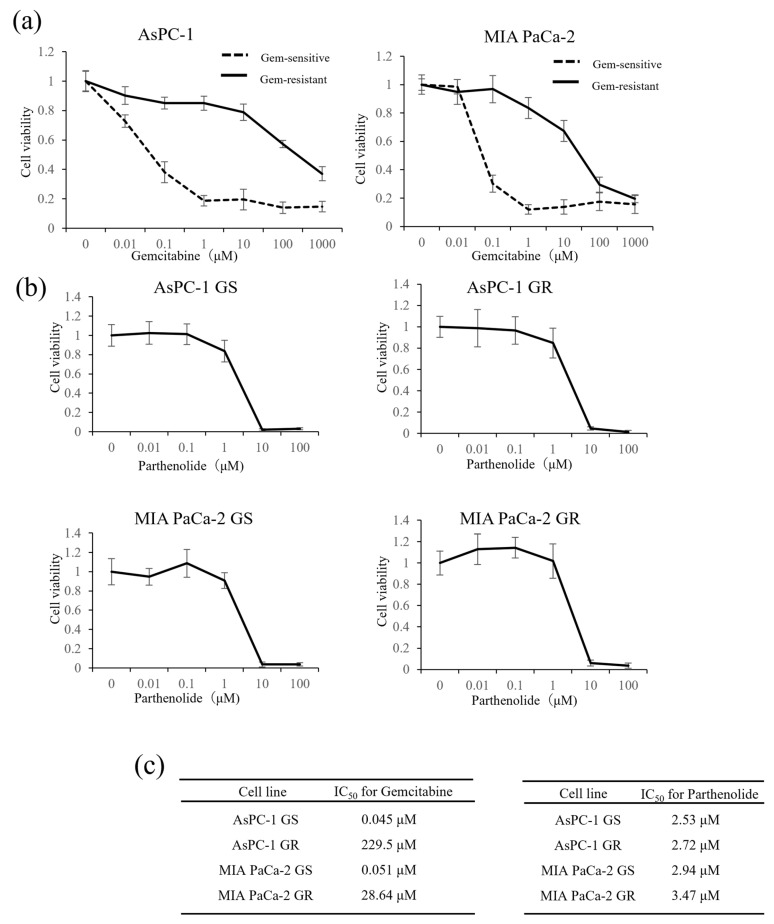
Effect of Gem on the proliferation of GS and GR PaCa cell lines and cytotoxic effects of parthenolide on GS and GR PaCa cell lines: (**a**) GS and GR AsPC-1 cells (**left**) or GS and GR MIA PaCa-2 cells (**right**) were treated with Gem at the indicated concentrations for 48 h, and the proliferation of each cell line was determined using a WST-1 assay. (**b**) PaCa cells (GS and GR AsPC-1 or GS and GR MIA PaCa-2) were treated with different concentrations of parthenolide for 48 h, and the viability of each cell line was assessed using a WST-1 assay. Values are expressed as the mean ± SD. (**c**) The IC_50_ of Gem in each PaCa cell line (**left**) and the IC_50_ of parthenolide in each PaCa cell line (**right**) were measured.

**Figure 2 nutrients-16-00705-f002:**
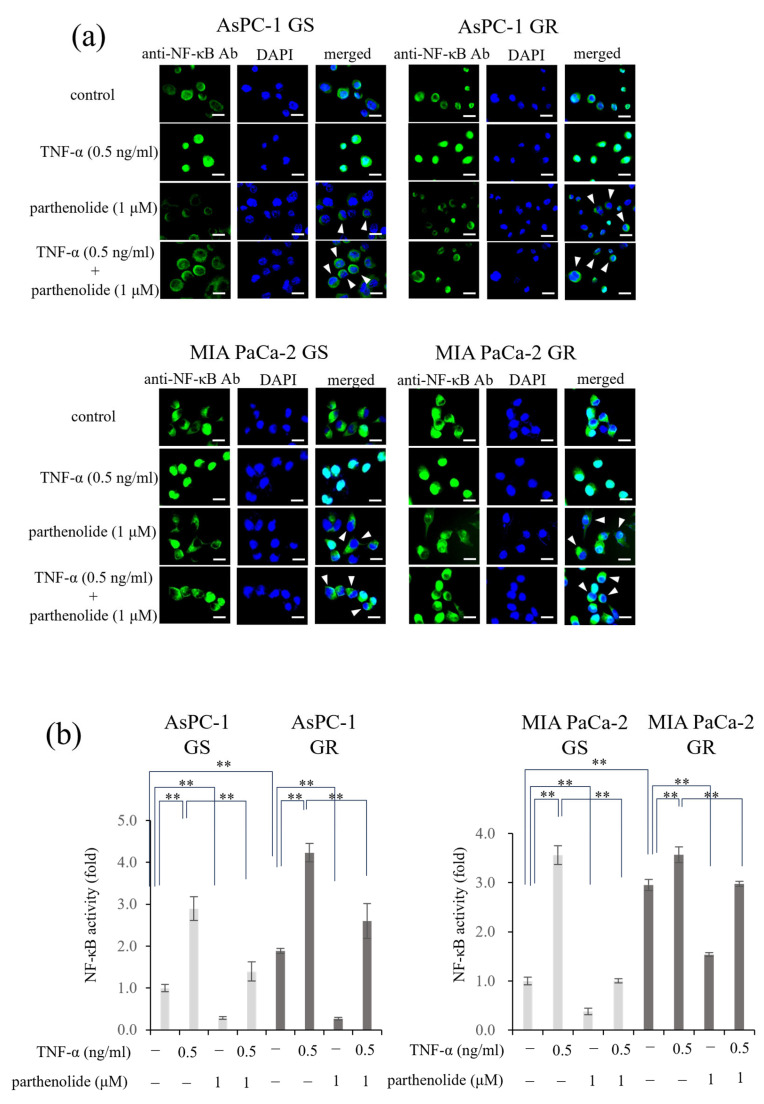
Effect of parthenolide on NF-κB activity in GS and GR PaCa cell lines: (**a**) Immunofluorescence staining was used to evaluate the nuclear translocation of p65. GS and GR AsPC-1 cells or GS and GR MIA PaCa-2 cells were treated with parthenolide (1 µM) and TNF-α (0.5 ng/mL). Representative images are shown (magnification, ×100; scale bar, 20 µm), and white arrow heads indicate cells in which the suppression of nuclear translocation was observed. (**b**) The activity of NF-κB in nuclear extracts was measured using a Trans AM NF-κB p65 kit. GS and GR AsPC-1 cells or GS and GR MIA PaCa-2 cells were treated with parthenolide (1 µM) and TNF-α (0.5 ng/mL). In both experiments, the treatment time for parthenolide was set to 1 h and that for TNF-α was set to 15 min before the end of incubation. Comparisons among groups were performed using one-way ANOVA with Bonferroni’s post hoc test. ** *p* < 0.05.

**Figure 3 nutrients-16-00705-f003:**
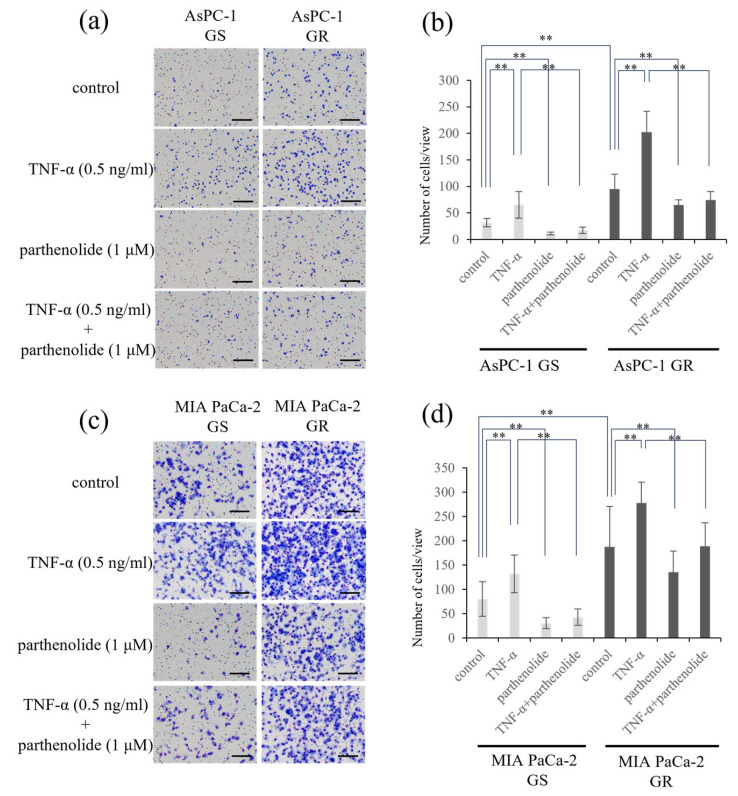
Effect of parthenolide on the invasiveness of GS and GR PaCa cell lines using Matrigel invasion assays. PaCa cells (1.0 × 10^5^ cells/well) were seeded in Transwell chambers pre-coated with Matrigel, and the PaCa cells were exposed to TNF-α (0.5 ng/mL) and/or parthenolide (1 µM). After 24 h incubation, the invaded cells were stained using Diff-Quik kit, and counted: (**a**) Representative images of GS and GR AsPC-1 cells after the indicated treatments are shown (magnification, ×40; scale bar, 200 µm). (**b**) The mean numbers of invaded cells (GS and GR AsPC-1) in nine random microscopic fields of view were compared among the treatments using a one-way analysis of variance with a post hoc Bonferroni test. ** *p* < 0.05. (**c**) Representative images of GS and GR MIA PaCa-2 cells after the indicated treatments are shown (magnification, ×40; scale bar, 200 µm). (**d**) The mean numbers of invasive cells (GS and GR MIA PaCa-2) in nine random microscopic fields of view were compared among the treatments using one-way ANOVA with Bonferroni’s post hoc test. Values are expressed as the mean ± SD. ** *p* < 0.05.

**Figure 4 nutrients-16-00705-f004:**
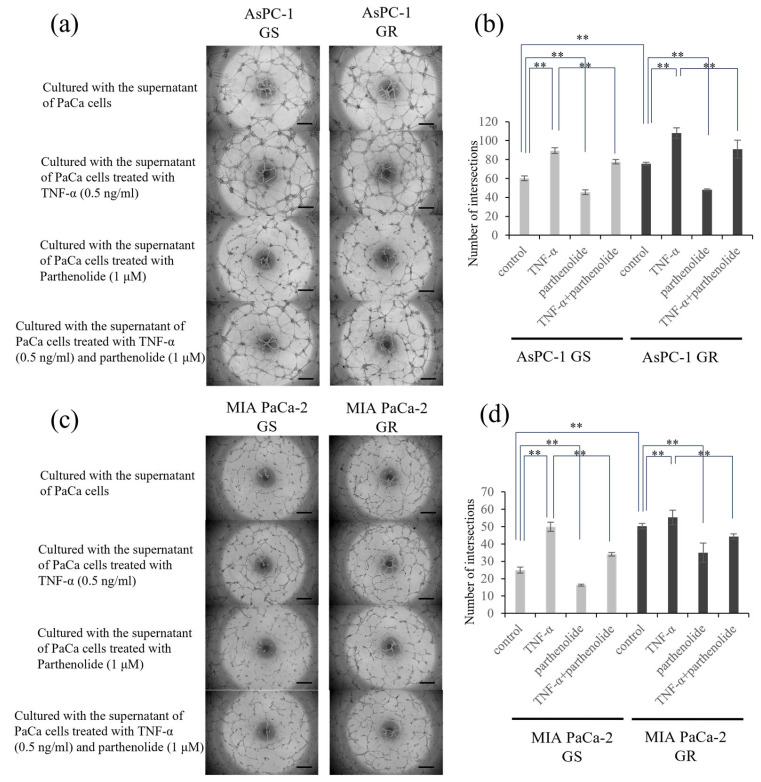
Effect of parthenolide on the angiogenesis of GS and GR PaCa cell lines. EA.hy926 cells (1.2 × 10^4^) were seeded on Matrigel and cultured with conditioned medium from PaCa cells treated with TNF-α (0.5 ng/mL) and/or parthenolide (1 µM): (**a**) Representative images of GS and GR AsPC-1 cells are shown (magnification, ×40; scale bar, 500 µm). (**b**) The number of intersections was counted and statistically compared among the different experimental groups of GS and GR AsPC-1 cells. ** *p* < 0.05. (**c**) Representative images of GS and GR MIA PaCa-2 cells are shown (magnification, ×40; scale bar, 500 µm). (**d**) The number of intersections was counted and statistically compared among the different experimental groups of GS and GR MIA PaCa-2 cells. Values are expressed as the mean ± SD. ** *p* < 0.05.

**Figure 5 nutrients-16-00705-f005:**
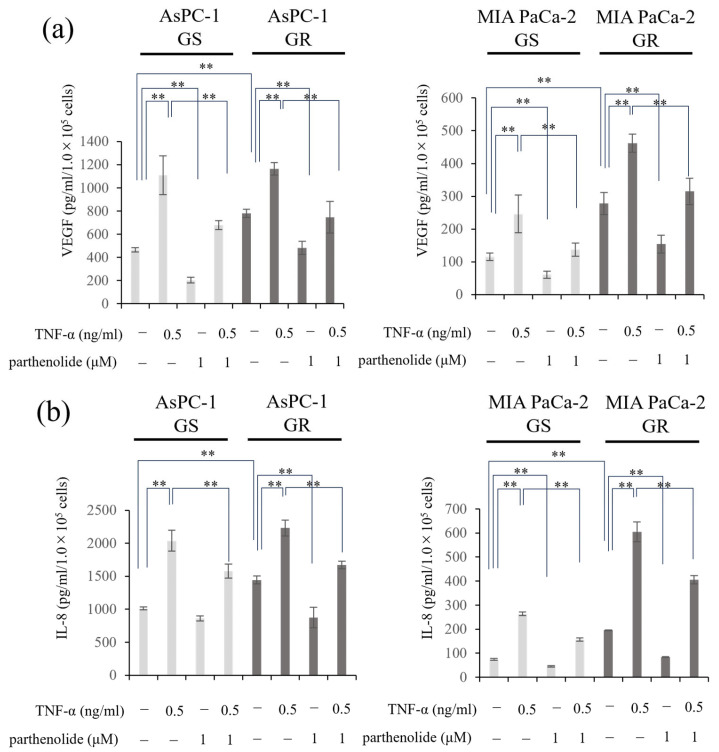
Effect of parthenolide and/or TNF-α treatment on VEGF and IL-8 secretion from GS and GR PaCa cell lines. PaCa cells (1 × 10^5^/well) were seeded in 6-well plates and incubated overnight. Then, the medium in each well was replaced with FBS-free medium with or without parthenolide (1 µM) and TNF-α (0.5 ng/mL), and the plates were incubated for 48 h. The concentration of the resulting cell supernatant was measured using ELISA: (**a**) VEGF secretion from GS and GR AsPC-1 cells or GS and GR MIA PaCa-2 cells treated with TNF-α and/or parthenolide are shown. (**b**) The changes in the IL-8 secretion from GS and GR AsPc1 cells or GS and GR MIA PaCa-2 cells treated with TNF-α and/or parthenolide are shown. Values are expressed as the mean ± SD. ** *p* < 0.05.

**Figure 6 nutrients-16-00705-f006:**
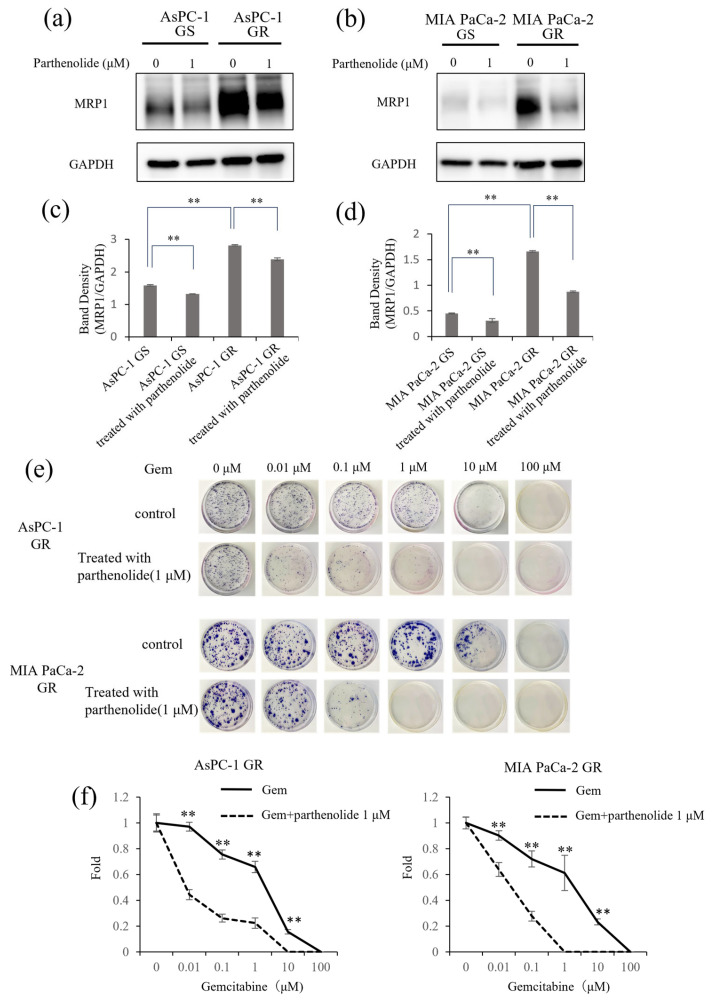
Effect of parthenolide on MRP1 expression in GS and GR PaCa cell lines. GS and GR AsPC-1 or GS and GR MIA PaCa-2 cell lysates were subjected to Western blotting to evaluate the effect of parthenolide on the expression of MRP1, an indicator of Gem resistance. Cells were initially treated with 1 µM parthenolide for 12 h. Then, cell lysates were harvested and subjected to Western blotting of MRP1. GAPDH was used as a loading control. The data are presented as the MRP1 level relative to the GAPDH level: (**a**,**b**) The protein level of MRP1 was evaluated using Western blotting. (**c**,**d**) The MRP1 level relative to the GAPDH level was assessed. Comparisons among groups were performed using one-way ANOVA with Bonferroni’s post hoc test. Data are presented as the mean ± SD. ** *p* < 0.05. (**e**) A colony formation assay showed that parthenolide improved the Gem sensitivity of GR AsPC-1 and MIA PaCa-2 cells. The experiments were performed in triplicate, and representative images are shown. (**f**) The number of colonies formed by GR AsPC-1 (**left**) and GR MIA PaCa-2 cells (**right**) were counted. Comparisons between Gem monotherapy and Gem-plus-parthenolide combination therapy were evaluated using a *t*-test. ** *p* < 0.05.

## Data Availability

The data presented in this study are available from the corresponding author upon request. The data are not publicly available due to privacy.

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
