# Peer review of "The Natural Product Parthenolide Inhibits Both Angiogenesis and Invasiveness and Improves Gemcitabine Resistance by Suppressing Nuclear Factor κB Activation in Pancreatic Cancer Cell Lines"

_nutrients, 2024, doi:10.3390/nu16050705_

Round 1

Reviewer 1 Report

Comments and Suggestions for Authors

The proposed article has special relevance since there are no published articles that study parthenolide on Gem-resistant pancreatic cancer cells.

The methodology is well described and detailed so that it can be replicated.

There are some aspects that can be improved:

- Figures 1 and 2 can be merged into a single figure. In the new figure, section A) could be maintained, where the two figures of the antiproliferative effect of Gem are shown. Next, section B) could be added with two figures of the antiproliferative effect of parthenolide, following the same style as section A, that is, Gem-R and Gem-S are represented in the same figure. Finally, in section C) the summary table of the Ic50 values ​​could be added.

- In section 3.2, it is necessary to check whether the reference of the figures corresponds to the text described or is the other way around, since when reference is made in the text, figure 3.A really refers to figure 3.B, and vice versa.

- In figure 3.B it is necessary to add the scale bar and indicate with arrows some nucleus where the translocation is observed.

- The results of section 3.3 should be more extensive and reference the sections of figure 4. In addition, it must include specific values ​​of the reduction percentage. Likewise, microscopy images must have the scale bar.

- In general, the results should be written more extensively and add specific reduction values...

- It is also necessary to reference more recent articles both in the introduction and in the discussion.

Comments on the Quality of English Language

Minor editing of English language required

Reviewer 2 Report

Comments and Suggestions for Authors

Manuscript nutrients-2863636

The natural product parthenolide inhibits both angiogenesis and invasiveness and improves gemcitabine resistance by suppressing nuclear factor‑κB activation in pancreatic cancer cell lines” for Nutrients

Comments:

1. Materials and methods: 2.2. Please provide the catalogue numbers of the applied cell cultures.

2. Materials and methods: 2.4. In the highest parthenolide concentration used (100 microM), the DMSO concentration is therefore 1%. Did the authors also check the toxicity of DMSO? Doesn't 1% have additive cytotoxic effects?

3. Materials and methods. 2.7. Why was the MRP1 protein chosen for evaluation but no other proteins from the ABC family? Others also participate in the acquisition of multidrug resistance by cells. The high amount of MRP1 in PaCa cells is not a sufficient explanation for merely assessing this protein from the entire family. Please explain this further and justify it better in the discussion.

4. Materials and methods. 2.11. Did the authors consider any specific number of cells that could be considered a colony? On what basis was it determined what is a colony and what is not?

5. Figure 3b. Please insert a scale bar at least in the control images.

Round 2

Reviewer 1 Report

Comments and Suggestions for Authors Icono de Validado por la comunidad The authors have correctly made the proposed changes.